# *azyx-1* is a new gene that overlaps with zyxin and affects its translation in *C. elegans*, impacting muscular integrity and locomotion

**Bhavesh S. Parmar[1], Amanda Kieswetter[1], Ellen Geens[1], Elke Vandewyer[1], Christina Ludwig[2], Liesbet Temmerman[1] \***

1 Animal Physiology and Neurobiology, University of Leuven (KU Leuven), Leuven, Belgium, 2 Bavarian Center for Biomolecular Mass Spectrometry (BayBioMS), Technische Universität München, München, Germany

\* Liesbet.Temmerman@kuleuven.be

**Data Availability Statement:** Mass spectrometry data are available via Panorama Public via https://panoramaweb.org/azyx1.url. All other relevant data

## Abstract

Overlapping genes are widely prevalent; however, their expression and consequences are poorly understood. Here, we describe and functionally characterize a novel *zyx-1* overlapping gene, *azyx-1*, with distinct regulatory functions in *Caenorhabditis elegans*. We observed conservation of alternative open reading frames (ORFs) overlapping the 5′ region of zyxin family members in several animal species, and find shared sites of *azyx-1* and zyxin proteoform expression in *C. elegans*. In line with a standard ribosome scanning model, our results support *cis* regulation of *zyx-1* long isoform(s) by upstream initiating *azyx-1a*. Moreover, we report on a rare observation of *trans* regulation of *zyx-1* by *azyx-1*, with evidence of increased ZYX-1 upon *azyx-1* overexpression. Our results suggest a dual role for *azyx-1* in influencing *zyx-1* proteoform heterogeneity and highlight its impact on *C. elegans* muscular integrity and locomotion.

## Introduction

Zyxins belong to a subfamily of conserved LIM domain-containing proteins found across eukaryotes and characterized for their role in cell-ECM (extra-cellular matrix) adhesion and cytoskeleton organization [1–4]. Characterized by a proline-rich N-terminus and 3 consecutive LIM domains in their C-terminal region [1,3,5], zyxins regulate actin assembly and remodeling, as well as cell motility [6–9]. In line with this, human zyxin is implicated in stretch-induced gene expression changes via active nuclear translocation [10]. Moreover, it also promotes apoptosis in response to DNA damage [11].

While multiple distinct zyxin proteins are present in vertebrates (for example, Lpp, Trip6, and Zyx), *Caenorhabditis elegans* contains a unique zyxin gene, *zyx-1*, with 5 annotated protein isoforms, of which isoforms a and b are predominantly expressed [5,12,13]. There are anatomical differences in isoform expression, with ZYX-1b observed in body wall muscle, pharynx, vulva, spermathecae, and multiple neurons, whereas ZYX-1a mainly localizes to tail phasmid neurons and uterine muscle, and weakly so, body wall muscle [13]. In *C. elegans*, *zyx-1* has been postulated to have a minor role in reproduction, however, the mechanism(s) and isoform

are within the paper and its Supporting Information files.

**Funding:** This work was supported by the Fonds Wetenschappelijk Onderzoek Flanders (FWO Flanders, grant G052217N to LT; www.fwo.be), the Katholieke Universiteit Leuven (KU Leuven, grant C16/19/003 to LT; www.kuleuven.be), and by EPIC-XS support to BSP, CL, LT through grant 823839 of the Horizon 2020 programme of the European Union (epic-xs.eu). Some strains were provided by the CGC, which is funded by NIH Office of Research Infrastructure Programs (P40 OD010440). The funders had no role in study design, data collection and analysis, decision to publish, or preparation of the manuscript.

**Competing interests:** The authors have declared that no competing interests exist.

**Abbreviations:** ECM, extra-cellular matrix; FA, formic acid; HDR, homology-directed repair; IRES, internal ribosome entry site; NGM, nematode growth medium; NMD, nonsense-mediated decay; ORF, open reading frame.

(s) involved remain elusive [14,15]. Beyond this, *C. elegans zyx-1* is hypothesized to be functionally analogous to vertebrate zyxin, with LIM domains acting as mechanical stabilizers at focal adhesions, and the proline-rich N-terminus involved in sensing muscle cell damage [12].

Previous studies revealed that only the ZYX-1b isoform regulates synapse maintenance and development, while in the context of a dystrophic mutant background, the longer ZYX-1a isoform partially rescues muscle degeneration in an ATN-1-dependent manner, highlighting how not only expression patterns, but also molecular functions are isoform-dependent for *C. elegans* zyxin [12,13]. At a gene-regulatory level, this use of alternative splicing products and functional diversification of proteoforms for *C. elegans* zyxin is in line with observations made for several LIM domain proteins across eukaryotes [16].

In general, alternative and overlapping open reading frames (ORFs) arising out of polycistronic mRNA can contribute to posttranscriptional regulation [17–19]. A recent community-wide effort for annotation of such genomic loci categorized overlapping genes based on their initiation and termination codon with respect to the main coding sequence of a given transcript [20]. Of these, ORFs with an upstream start site (upstream or upstream overlapping; uORF and uoORF) often influence the translation of the main coding sequence, based on the evidence of the ones that have been detected and investigated in detail [21–25]. From a more human-centered future perspective, uORFs are a rather unexplored niche for translational research: with a predicted prevalence in over 50% of human genes and first examples regulating translation of disease-associated genes already emerging [22,26], the field is bound to not only lead to more fundamental, but also application-oriented insights. Keeping this broader context in mind, we here focus on more fundamental principles of uORFs in a model organism context.

We previously provided mass spectrometric evidence for 467 splice variants and 85 noncanonical gene products, including from polycistronic and ncRNA translation, in *C. elegans* [27]. Of these, 1 newly discovered gene, *azyx-1* (alternative N-terminal ORF of *zyx-1*), was identified as an 166 amino acid-long protein, translated from the 5′ UTR of *zyx-1*. In this study, we provide evidence for 2 protein isoforms of *azyx-1*, one initiating upstream and another downstream of *zyx-1* AUG, and both overlapping the proline-rich N-terminus of ZYX-1 long isoforms. To understand whether zyxin proteins could also be regulated by upstream/overlapping ORFs, we here explore functional relevance of *azyx-1* and its relation to zyxin in the model organism *C. elegans*.

## Results

### A novel gene, *azyx-1*, reveals putative syntenic conservation of overlapping genes on eukaryotic zyxin

The *C. elegans* genome contains 2,468 predicted ORFs initiating in 5′ UTRs; however, only 8 of those are supported by mass spectrometric evidence [27,28]. One of these is *azyx-1*, a noncanonical gene overlapping the gene encoding the only *C. elegans* zyxin, *zyx-1* [27]. *azyx-1* is translated from a different reading frame than *zyx-1*, starting from an AUG 184 bp upstream of the *zyx-1* initiation site and delivering a polypeptide product with a sequence length of 166 amino acids (AZYX-1a, Fig 1A). The same locus also embeds a putative shorter AZYX-1 isoform (AZYX-1b), initiating within the *zyx-1* coding sequence and containing only the 106 C-terminal amino acids of AZYX-1a (Fig 1A). Six different transcripts have been described for the *zyx-1* locus (F24G4.3, WormBase version WS283): 2 are associated with isoform a, and then a single transcript for isoforms b-e each (Fig 1A). Based on these, AZYX-1a can be translated from the F42G4.3a.1 transcript that contains a longer 5′ UTR, whereas AZYX-1b could be translated from all 3 transcripts of the longer *zyxin* isoforms a and e: F42G4.3a.1, F42G4.3a.2, and F42G4.3e.1. We previously mapped unique peptides to *azyx-1*, covering

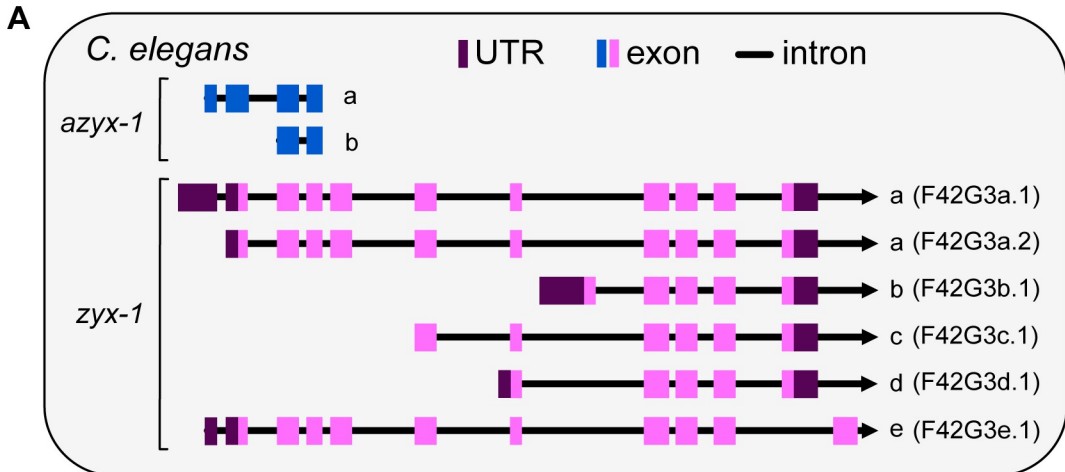

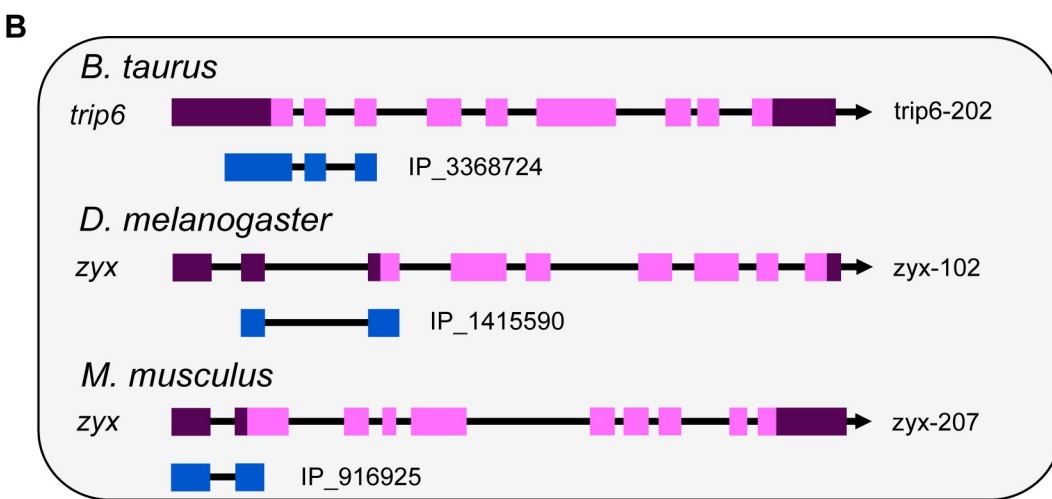

**Fig 1. Schematic representation of zyxin loci including overlapping uoORFs.** Exons (pink: canonical, blue: uoORF/oORF), introns (line), and UTRs (purple) indicated. All loci are shown according to the direction of transcription (arrowhead). (**A**) Six zyx-1 transcripts are recognized in *C. elegans*, corresponding to 5 isoforms. The coding sequence for the uoORF/oORF (blue) azyx-1 as identified previously by Parmar and colleagues is contained in the long transcripts. (**B**) Examples of uoORFs (blue) predicted by OpenProt [28] with respect to zyxin orthologs of *B. taurus*, *D. melanogaster*, and *M. musculus*. The gene names (left), transcript ID (right), and OpenProt IDs are provided next to each schematic representation.

>60% of the 166 amino acids-long protein sequence at 1% FDR [27], showing that *azyx-1* indeed is translated. Since LIM-domain proteins, to which zyxins belong, often result from alternative splicing and display strong domain conservation across eukaryotes [16], we asked whether *azyx-1* or the phenomenon of overlapping genes at zyxin loci is also conserved across eukaryotes. To that end, we searched for predicted ORFs initiating in 5′ UTRs (uORF and uoORF) or within the coding sequence (oORF) of zyxin orthologs (*lpp*, *trip6*, and *zyx*) in 9 animals (OpenProt release 1.6, S4 Table). We identified 14 ORFs initiating upstream of *zyx-1* orthologs in 7 species including *C. elegans* (Fig 1B and S4 Table). Moreover, 6 of these 5′ UTR initiating ORFs were conserved in vertebrates as alternative oORFs within the coding region of zyxin orthologs (S4 Table). Together, our observations suggest a syntenic conservation of overlapping genes towards the 5′ end of zyxin orthologs, in combination with the widely prevalent alternative splicing observed in zyxins.

### *azyx-1* and *zyx-1a* reporters are observed in partially overlapping anatomical locations

To understand which cells or tissues of *C. elegans* may express *azyx-1*, we generated an extra-chromosomal *azyx-1* reporter strain (LSC1959). While the *zyx-1*a start codon is contained within this sequence, it does not share the *azyx-1* reading frame. Hence, the *azyx-1::mNeon-Green* fusion construct cannot lead to a fluorescent signal should translation initiate at the downstream *zyx-1*a start codon. We observed a strong fluorescent signal in body wall muscle, vulval muscle, and very faintly in unidentified structures in the head and tail (Fig 2A–2D). This fluorescence pattern was consistent, as observed in L4 and adult worms (Figs 2 and S1A–S1D). Next, we looked for expression of *zyx-1* isoforms as reported via extrachromosomal arrays by Luo and colleagues. In line with their report, we also observe the *zyx-1* long isoform predominantly in tail neurons, and faintly in body wall muscle, and additionally also faintly in the pharynx, and more brightly in an unidentified structure in the head (Fig 4A, middle panel of control). Interestingly, the short *zyx-1b* isoform is also strongly expressed in body wall and vulval muscle [13] (Luo and colleagues), where we observe *azyx-1* expression (Fig 2A). Thus, beyond alternative splicing in zyxins, our observations suggest that in those locations, proteoform diversification and localization fine-tuning may putatively also occur via polycistronic transcripts, with the observed expression patterns suggesting the possibility of mutually exclusive translation from long *zyx* transcripts.

### AZYX-1 and ZYX-1 levels vary with age

To understand which protein products are generated by these overlapping genes, we quantified AZYX-1 and ZYX-1 at 3 different ages: L4 larvae, and young (day 1) and post-reproductive (day 8) adults. We selected these life stages based on *azyx-1* and *zyx-1* expression in body

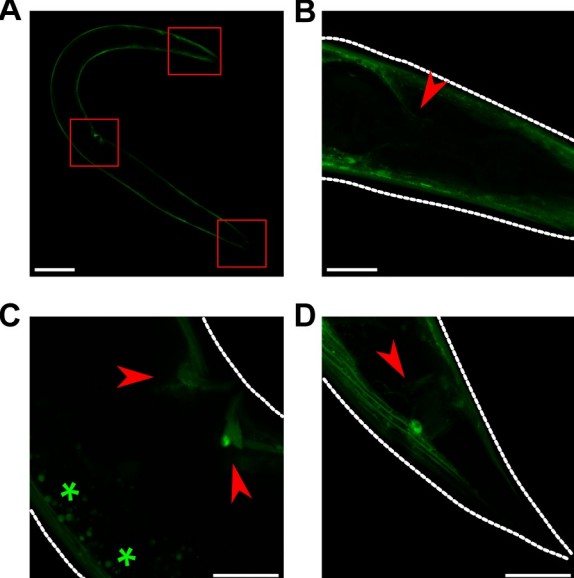

**Fig 2. azyx-1 is prominently expressed in body wall and vulval muscle and faintly in the head and tail region.** Anterior always to the left. (**A–D**) For azyx-1 localization, azyx-1p::azyx-1::mNeonGreen:: azyx-1 3′ UTR was expressed extra-chromosomally in a wild-type background. The reporter protein is clearly visible in (**A**) body wall muscle (scale bar, 100 μm), with red boxes indicating (**B**) 2 unidentified neurites in the head, (**C**) vulval muscle, and (**D**) in the tail region (scale bar, 20 μm, autofluorescence *). For L4 life stage, see S1A–S1C Fig.

wall muscle (Fig 2), combined with the knowledge that *C. elegans* musculature deteriorates with age [29]. We observed a relative increase of about 4-fold for AZYX-1 ($p = 8.3 \times 10^{-5}$) and 6-fold for ZYX-1 ($p = 1.9 \times 10^{-5}$) in day 1 adult versus L4 stages (Figs 3A and S2). While AZYX-1 at day 8 of adulthood remained high ($p = 0.77$, Figs 3A and S2A), ZYX-1 of these post-reproductive animals was intermediary between L4 and day 1 adult levels (Fig 3A). By relying on quantitative data of the specific peptides making up these proteins, we can further deduce that changes in adult zyxin levels differ for different proteoforms: at day 8 versus day 1 of adulthood, 2 out of 3 quantified N-terminal peptides (ZYX-1.1 and 1.3) were comparable to day 1 levels, while the quantified C-terminal peptides (ZYX-1.4 and ZYX-1.5) significantly reduced ($p = <0.02$, Figs 3B and S2B). For AZYX-1, 5 out of 7 peptides remain unchanged while 2 (shared by both AZYX-1a and AZYX-1b) show a decline (Fig 3C). Together, these data suggest that AZYX-1 levels remain stable while ZYX-1 levels decline between day 1 and day 8 of adulthood. This is corroborated by similar transcript level fold change of *zyx-1* for day 1 and day 8 adulthood [30]. Furthermore, it appears that shorter isoforms of *zyx-1* may reduce with age, whereas the longer isoforms might be less susceptible to such a post-reproductive decline.

### *azyx-1* likely exhibits *cis* control over *zyx-1*

Since *azyx-1* initiation lies 184 bp upstream of *zyx-1*, we hypothesized a *cis* regulation of downstream *zyx-1* major ORF by *azyx-1*. To test whether this could be the case, we generated an *azyx-1a* deletion mutant (LSC1898, Δ27bp of *azyx-1a*) and quantified ZYX-1 peptides. We observed a substantial increase in N-terminal ZYX-1 peptides (ZYX-1.1–1.3) for *azyx-1* mutants versus wild type (Fig 3D). Peptides that were not specific to the long zyxin proteoforms (ZYX-1.4–1.6) showed a much more modest change—or even none at all—in mutant versus control. The N-terminal peptides, specific to *zyx-1* long isoforms (ZYX-1a/e), showed a 2- to 3-fold range increase upon *azyx-1a* mutation. To confirm that the observations for the deletion mutant most likely result from loss of ribosomal initiation at the AZYX-1a start site, we generated a second mutant through precise nucleotide exchange to mutate it from ATG to TAC (PHX7030). Indeed, both mutants similarly increase ZYX-1$_{long}$ peptides while leaving the others unaffected (S4A Fig). Our observations also hold true for all 3 independent means of data normalization that were available (GDP-3, spike-in, HIS-24; Figs 3D, S3A, S3B, and S3D), supporting the solidity of the claim that removal of the *azyx-1a* start codon increases zyxin levels in a proteoform-biased way. AZYX-1a mutants also suppressed all downstream peptides shared by AZYX-1b, with several remaining below detection limit in mutants. This suggests that AZYX-1a might be the prominently translated isoform (S3C and S4B Figs). Together, our results fit the hypothesis that *azyx-1* may inhibit translation of downstream *zyx-1* isoforms, likely affecting the longer isoforms more due to a shared transcript. This does not rule out the possibility of *trans* regulation (see below) or of an internal ribosome entry site (IRES) downstream of the AZYX-1a start codon for ZYX-1a initiation; however, based on in silico analysis, there is no evidence of IRES within the region downstream of the AZYX-1a start site.

### *azyx-1* overexpression increases ZYX-1 levels and accumulation in GABAergic motor neurons

While we suspect that uoORF *azyx-1a* may exhibit *cis* control over translation of *zyx-1* long isoforms, measurable quantities of AZYX-1 peptides are being produced in vivo (Figs 2 and 3, Parmar et al. 2021 [27]). Given the detection of two isoform-specific peptides (S2 Table), these certainly are translation products of *azyx-1a*. A putative shorter isoform, *azyx-1b*, with a predicted start codon downstream of that of the *zyx-1* long isoforms could also contribute to the

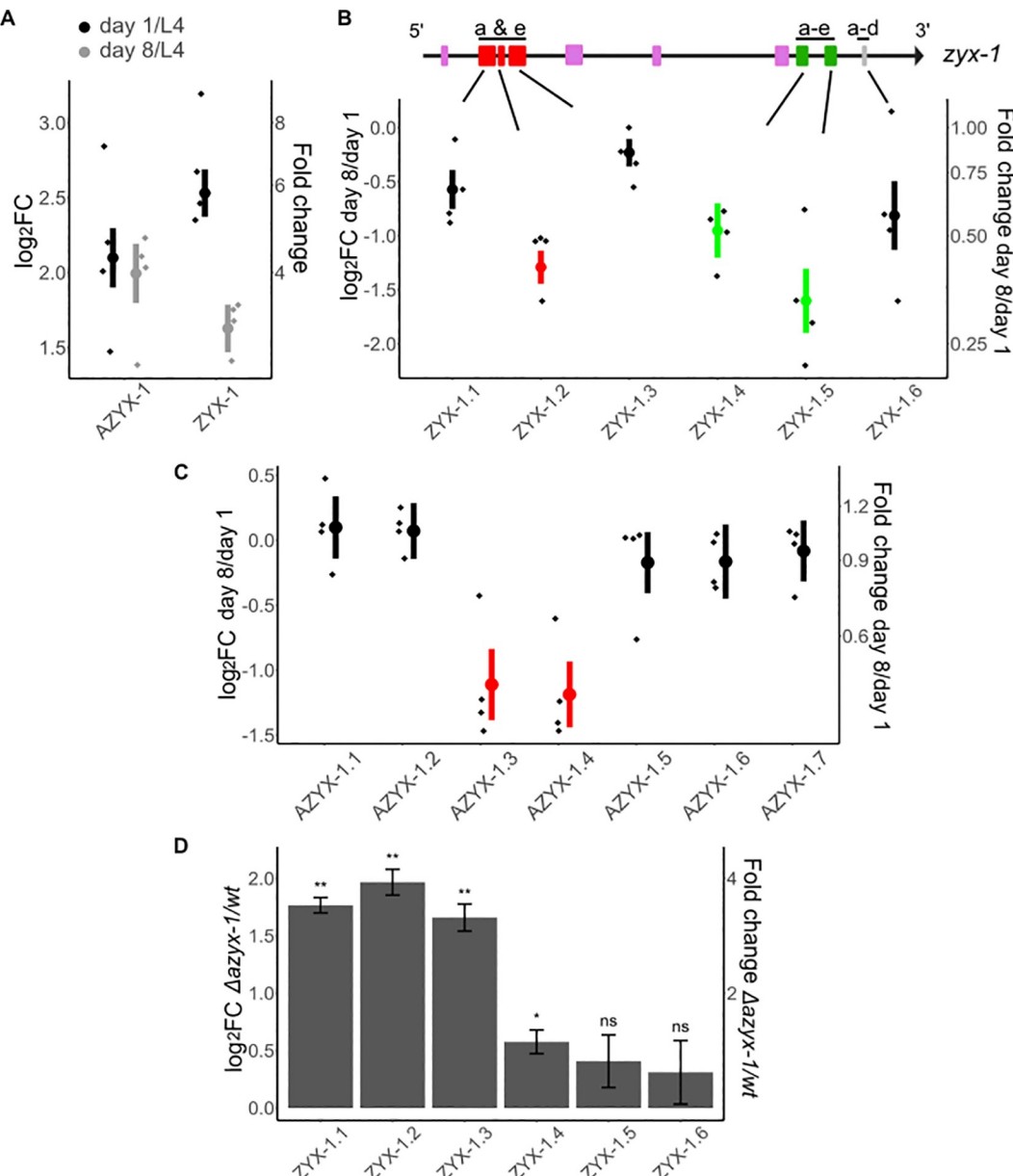

**Fig 3. Targeted quantitation in aging worms and in an azyx-1a deletion mutant reveals age-dependent zyx-1 fold change and *cis* regulation by azyx-1.** Graph axes indicate $\log_2$ fold change (left) and absolute fold change (right). Data normalized to a spike-in peptide (1 fmol/worm) for aging wt worms and GPD-3 for mutant vs. wt comparison, with 4 biological replicates (diamonds) for each time point, mean (circle), and standard error bars. (**A**) Fold change at day 1 (black) and day 8 (gray) of adulthood in comparison to L4 larval stage for all quantified peptides of AZYX-1 (7 peptides) and ZYX-1 (6 peptides) combined. (**B**) Fold change at day 8 in comparison to day 1 of adulthood for individual ZYX-1 peptides (1.1 to 1.6). zyx-1 gene model indicates the location of peptides unique to long isoform a and e (1.1–1.3) and peptides shared by longer and shorter isoforms (1.4–1.6). (**C**) Fold change at day 8 in comparison to day 1 of adulthood for individual AZYX-1 peptides (1.1 to 1.7). AZYX-1.1 and 1.2 are unique to AZYX-1a while the rest are shared by AZYX-1a and AZYX-1b. Significantly decreased peptides are colored as per their position relative to the overlapping zyxin gene model (red: long, green: shared; fold change cut-off = <0.66, *p*-value = <0.02). (**D**) Peptides unique to ZYX-1 long isoforms (ZYX-1.1–1.3) show a consistent increase in the azyx-1 deletion mutant, while peptides shared between all ZYX-1 (1.4–1.6) isoforms remain similar to wild-type levels (at day 1 of adulthood); p-value: * <0.05, ** <0.01, *** <0.001, ns = not significant, *n* = 4 biological replicates, with data normalized to GPD-3. For spike-in and HIS-24 normalization, see S3A and S3B Fig. Data used to generate figures can be found in S1 Data.

remainder of the measured signal (Figs 1A, 2, and 3; Parmar and colleagues [27]). Therefore, we asked whether *azyx-1* might also influence *C. elegans* zyxin in *trans*. We made use of a reporter system that fuses all zyxin isoforms to GFP and only the long ones (a/e) additionally to mCherry [13] and extrachromosomally expressed *azyx-1* (Fig 4A). We measured a significant increase in all (*zyx-1::GFP*), as well as in the long (*mCherry::zyx-1*) proteoform reporter levels specifically, upon *azyx-1* overexpression in the reporter strain ($p < 0.001$; Fig 4B and 4C). This observation suggests that production of ZYX-1 may be stimulated by the presence of AZYX-1. In addition, the ratios for mCherry/GFP show no significant differences, suggesting an overall ZYX-1 increase under forced overexpression of *azyx-1*, irrespective of proteoform (Fig 4D). So far, our results support the presence of both *cis* and *trans* control of *zyx-1* by *azyx-1* in vivo.

Upon *azyx-1* overexpression, we also observed an increase in GFP signal aggregation along the ventral nerve cord in what appeared to be motor neurons (Fig 4E). Colocalization with *unc-47p::mCherry*-positive cells identifies GABAergic neurons along the ventral nerve cord as sites of this *zyx-1* accumulation under *azyx-1* overexpression with an observed increase in GFP signal (Fig 4F and 4G). *zyx-1* has previously been reported to be expressed in GABAergic neurons [31], and its accumulation in these motor neurons upon increased AZYX-1 levels made us hypothesize that *azyx-1* overexpressors may suffer locomotion-related impairments.

## *azyx-1* overexpression affects muscle integrity and neuromuscular behavior in *C. elegans*

Our expression analysis revealed that *azyx-1* and *zyx-1* are abundantly expressed in body wall muscle (Fig 2A and S1). In line with this, using blinded and randomized manual scoring (Fig 5A), we observed a quantifiable loss of muscle fiber integrity in *azyx-1* overexpression conditions but not in deletion mutants, as compared to control animals (Fig 5B; Fisher's exact test: wild type versus OE line 1 $p = 4.13 \times 10^{-4}$, versus OE line 2 $p = 0.0287$, versus mutant $p = 1$). To test whether manipulating AZYX-1 levels could, as is already known for *zyxin* deletion [12], affect muscle performance, we performed burrowing assays. These evaluate the ability of worms crawling through Pluronic F-127, a transparent and biocompatible gel, stimulated by a chemoattractant, giving insight into the neuromuscular capabilities of the animals [32]. Here, the genetic overexpression of *azyx-1* resulted in defective burrowing as compared to wild type (Fig 5C, two-way ANOVA, Tukey HSD wild type versus *azyx-1* OE line 1 $p = 3.14 \times 10^{-6}$, versus *azyx-1* OE line 2 $p = 4.35 \times 10^{-10}$; S5 Fig, versus *azyx-1* OE line 3 $p = 0.046$), while *azyx-1* mutants displayed contrasting results. The Δ27bp mutant displayed borderline-significant defective burrowing (S5 Fig, $p = 0.047$) that could be rescued by extrachromosomal resupplementation of *azyx-1* (S5 Fig, $p = 0.401$), but mutating the *azyx-1*a start site did not affect burrowing capabilities (Fig 5C, $p = 1$). Together, these data suggest that abnormally high doses of AZYX-1 affect muscular integrity and neuromuscular behavior.

## Discussion

Although noncanonical ORFs are widely prevalent in eukaryotic genomes [33–35], they often tend to be species specific [18,36,37]. These ghost genes often are not susceptible to BLAST analysis due to their small size and inherent genomic variation over species. Upon manually examining the genomic loci of zyxin orthologs, we found evidence of putative syntenic conservation of *azyx-1* across 7 species (Fig 2B and S4 Table). This exemplifies how noncanonical ORFs can escape functional annotation, as the existing automated means for orthology mapping are restricted to (large) sequence similarity. Naturally, overlapping genes are more likely to be conserved across species, as a consequence of parent gene conservation. It is interesting

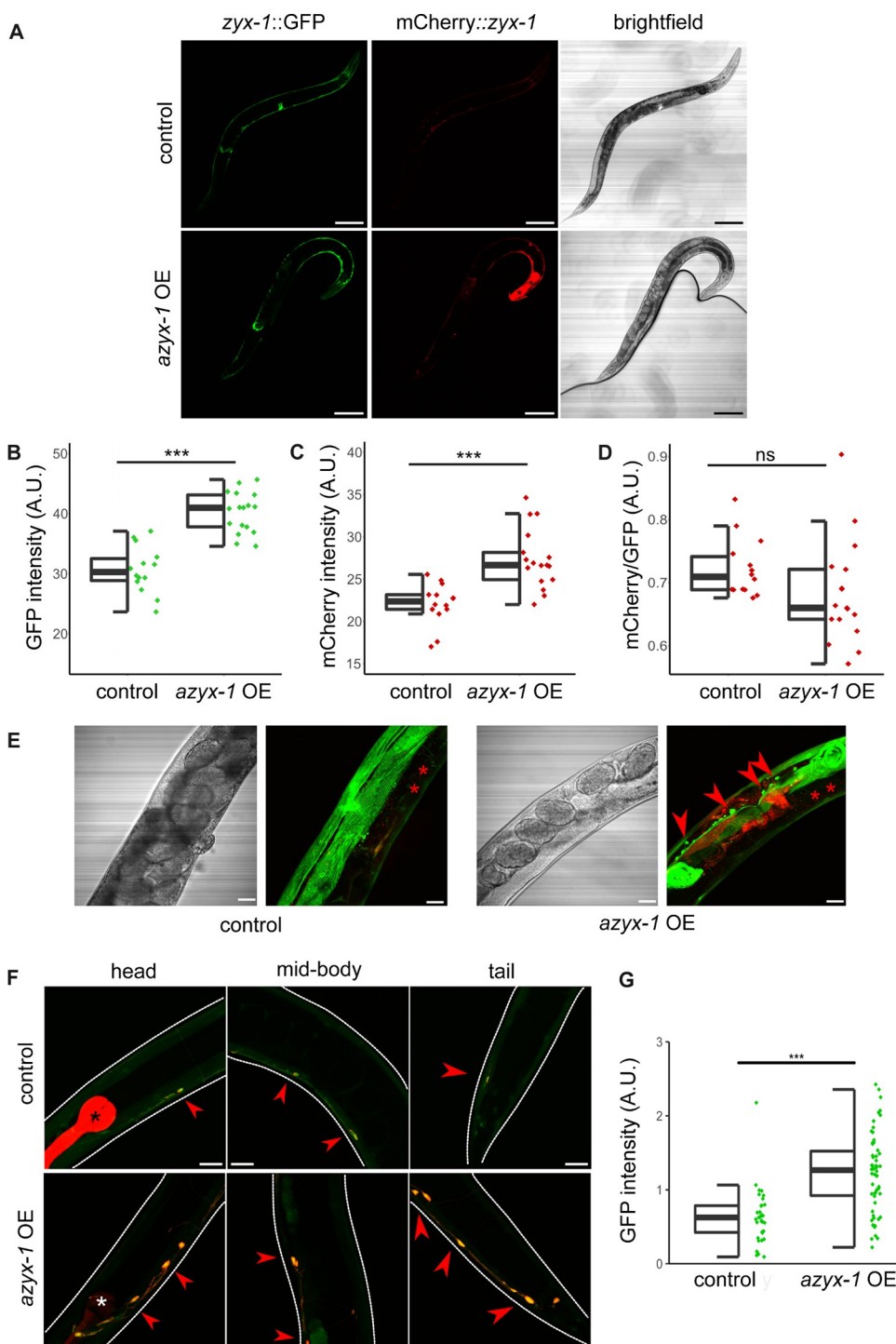

**Fig 4. azyx-1 overexpression increases zyxin reporter signal and leads to zyxin accumulation in motor neurons.**
(**A**) The integrated reporter control (LSC1870) (upper) contains ZYX-1::GFP (representing all isoforms) and mCherry::ZYX-1 (representing long isoforms), which is compared against a strain with the same genetic reporter background (lower), extrachromosomally overexpressing azyx-1 (azyx-1OE, LSC1960). Anterior to the right; scale bar, 100 μm. (**B**) GFP ($p = 4.85 \times 10^{-8}$) and (**C**) mCherry ($p = 4.82 \times 10^{-5}$) quantification showing increase in signal upon azyx-1 overexpression, with * indicating significance (ANOVA) for $n \geq 14$ per condition. (**D**) Ratio of mCherry/GFP. (**E**) Increased accumulation of GFP along the ventral nerve cord as observed in azyx-1OE (LSC1960) compared to control reporter (LSC1870), *intestinal autofluorescence and (**F**) colocalization (merged mCherry and GFP, red arrows) of zyx-1::GFP in motor neurons upon azyx-1 overexpression as imaged in adult head, mid-body, and tail. Upper panels: control worms (LSC1998) only expressing the neuronal reporter construct (unc-47p::mCherry) and *co-

injection marker (myo-2p::mCherry), lower panels: azyx-1 OE (LSC1999) in the control background. Scale bars: 20 μm. (**G**) GFP quantification in unc-47p::mCherry-positive cells showing increase in signal upon azyx-1 overexpression (Welch's *t* test $p = 2.25 \times 10^{-11}$, $n \geq 35$ neurons per condition). Data used to generate figures can be found in S1 Data.

however, that overlapping genes could also be harbored within specific gene isoforms to potentially regulate gene expression at a proteoform level, which we propose is the case for *azyx-1a* and *zyx-1a*.

Since *azyx-1a* and *zyx-1a* likely share a transcript, we asked whether there might be a preference for translation initiation from either of the 2 start codons. Upon *azyx-1a* start codon

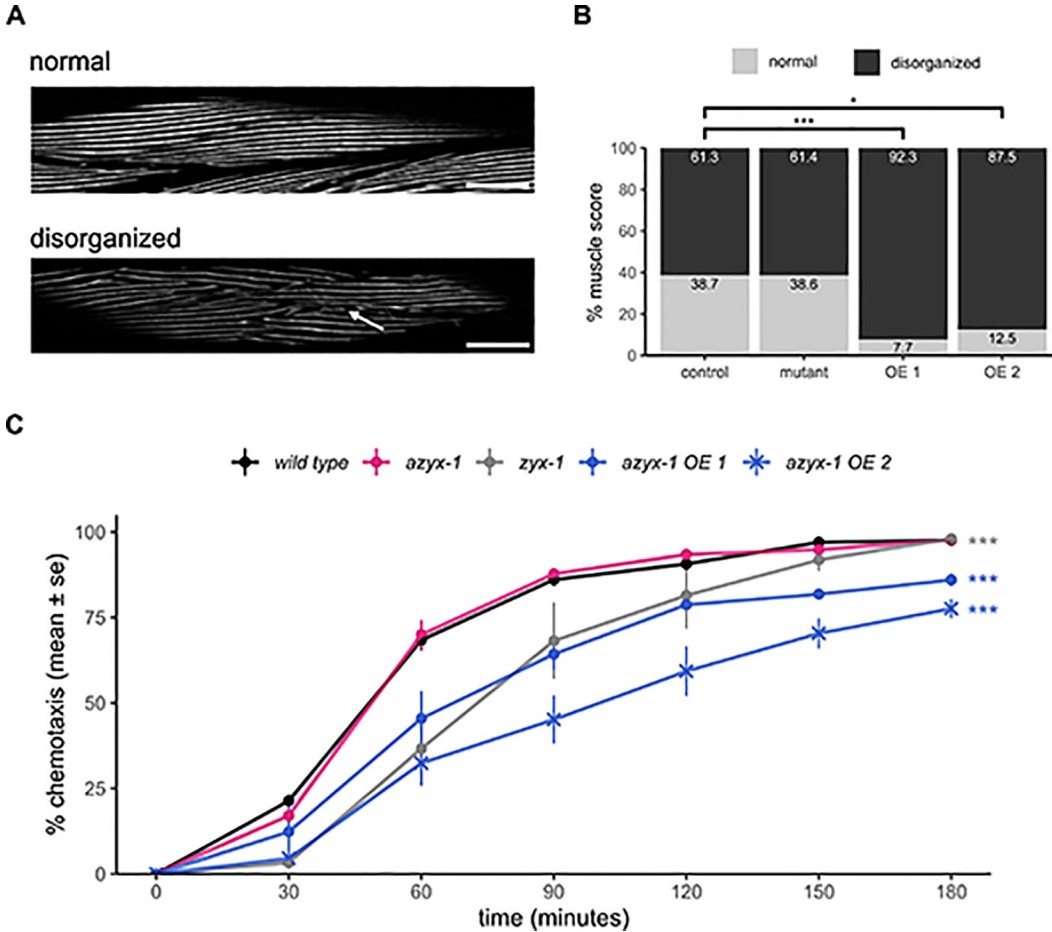

**Fig 5. azyx-1 overexpression affects muscle integrity and burrowing behavior.** A fluorescent myo-3 reporter (RW1596) was either crossed with an azyx-1 deletion mutant (LSC2001) or injected to create azyx-1 overexpressor strains (LSC2000, LSC2055) to score muscular quality of day 1 adults via blinded image analysis ($n \geq 40$ per condition). (**A**) Images were scored as disorganized when differing from controls at least as severely as the example image shown here, deteriorated region indicated with arrow. (**B**) Distribution of observed muscular phenotypes in azyx-1 mutant and overexpressor (OE) lines as compared to controls; Fisher's exact test $p = 1.483 \times 10^{-5}$, pairwise wild type vs. OE line 1 $p = 4.13 \times 10^{-4}$, vs. OE line 2 $p = 0.0287$. (**C**) Chemotaxis index of burrowing assay in day 1 adults, as cumulatively observed over 180 min for the azyx-1 ATG-to-TAC mutant (PHX7030) and for overexpressor lines (line 1: LSC2052, line 2: LSC2053) in comparison to positive (zyx-1(gk190)) and negative (wild type) controls, $n \geq 30$ per condition in 4 biologically independent replicates ($\geq 30 \times 4$); *p*-value two-way ANOVA $2 \times 10^{-16}$ for strain and time. Tukey HSD wild type vs. azyx-1 OE line 1 $p = 3.14 \times 10^{-6}$, vs. azyx-1 OE line 2 $p = 4.35 \times 10^{-10}$, vs. zyx-1 $p = 4.51 \times 10^{-8}$. Data used to generate figures can be found in S1 Data.

mutation, we observed a substantial increase in ZYX-1 long isoform abundance, while the downstream short isoforms largely remain unaffected (Figs 3D, S3A and S3B). Given the global linearity of eukaryotic ribosomal scanning and translation [38,39], our results suggest that *azyx-1a* exhibits *cis* control over *zyx-1* long isoform(s), by occupying the scanning ribosome in +2 reading frame. While we cannot fully exclude a direct effect on *zyx-1a* transcription due to a possibly nested effect of any *azyx-1* mutation with changes to a putatively unknown regulatory element of the *zyxin* gene, our observations in 2 independent mutants are in coherence with the widespread *cis* repression of downstream major ORF translation by upstream ORFs, predicted across vertebrates and observed in cells and animal models [40–42].

Interestingly, *azyx-1* also contains a downstream oORF isoform, prompting a putative *trans* function. One option would be for AZYX-1 to drag the remaining transcript to nonsense-mediated decay (NMD), as has been observed for other uoORFs in plants, yeast, and mammalian cells [43–47]. However, *zyx-1* is not present in the available *C. elegans* resource for NMD targets [48] and contrary to such expectations, we observe a positive *trans* effect of *azyx-1* on *zyx-1* reporter constructs (Fig 4). We therefore propose there may be a feed-forward loop that regulates *zyx-1* (long and short) translation via *azyx-1* (Fig 6). This is in any case a rare observation of reinforcing *trans* control by an upstream (overlapping) gene, which based on our observations is opposed to its proposed uoORF-mediated *cis* effect (Fig 3D). A similar observation of a combined *cis* and *trans* regulation has been made before for human hepatitis B virus [49], and it is possible that the regulatory context could be proteoform specific. While the *cis* regulation can be explained by standard ribosomal scanning, the molecular interplay involved in sensing AZYX-1 to then regulate *zyx-1* in *trans* remains to be investigated. Since *azyx-1* extends 2,475 bp into the *zyx-1 long* ORF, it is unlikely for ribosomes to reinitiate at *zyx-1 long* start codons after being released from the *azyx-1* termination site. We hypothesize that the *zyx-1*-overlapping C-terminus of *azyx-1* contains a functional *trans* domain responsible for the feed-forward loop. Previous studies have shown physical interaction between overlapping genes [50,51], but the mechanisms by which this occurs often remain undiscovered. Our results suggest that *azyx-1/zyx-1* is an interesting candidate to elucidate such interactions and intragenic *trans* regulation.

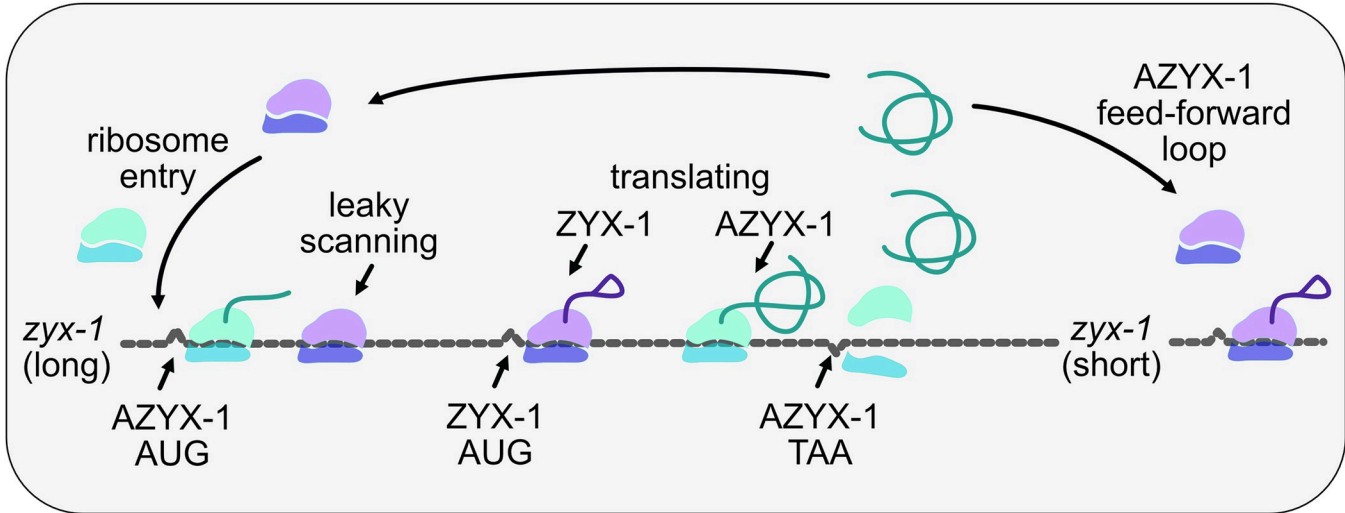

**Fig 6. Proposed model of *cis* and *trans* regulation of *zyx-1* by *azyx-1*.** Ribosomes upon entry onto zyx-1 (long) mRNA initiate translation (green) at uoORF AZYX-1a AUG in +2 reading frame, thus repressing translation of downstream ZYX-1 in *cis*. Leaky scanning through upstream AZYX-1a initiation leads to ZYX-1 translation (pink). Accumulation of translated AZYX-1 causes increased translation of zyx-1 (long and short) mRNA in a feed-forward loop via an unknown signaling cascade.

Phenotypically, increased localization of *zyx-1* in motor neurons (Fig 4F and 4G), defective muscle spindles, and burrowing chemotaxis upon *azyx-1* overexpression (Fig 5) suggest *azyx-1* as a key player in *zyx-1*-mediated muscular integrity and locomotion. Since *zyx-1* shorter isoforms (b and d) are mainly expressed in body wall muscle and motor neurons [12,13], we anticipate that the observed increase upon *azyx-1* overexpression corresponds to shorter and not *zyx-1* long isoforms (Fig 5B). Because long zyxin transcripts (which also contain *azyx-1*) can be detected in different anatomical sites (Figs 2 and S1A–S1C; Lecroisey and colleagues and Luo and colleagues [12,13]), it will be interesting to, in future experiments, resolve how all observed effects may depend on cell- or tissue-dictated differences in *(a)zyx-1* transcription and translation.

Previous studies observed substantial decrease in worm muscle degeneration upon RNAi knockdown of 5′ region of *zyx-1a* and 3′ of *zyx-1* in dystrophic background (*dyc-1; hlh-1*), with a distinct isoform involvement [12]. Our results show that *azyx-1* overexpression leads to *zyx-1* increase and muscle deterioration, while a *azyx-1* mutants have significantly increased *zyx-1* long isoforms and do not suffer measurable muscular defects (Fig 5B). We hypothesize that the observed effect stems from 2 distinct pathways, which may or may not involve direct modulation by *azyx-1*. The *zyx-1 long* isoform, with its characteristic N-terminal proline rich region, has been hypothesized to translocate to the nucleus under cytoplasmic stress and activate transcription of genes involved in repair of damage [12]. This would explain the muscular integrity of *azyx-1* mutants, possibly driven by their *zyx-1* long isoform increase. On the other hand, *azyx-1* overexpression could be causing an imbalance in long/short *zyx-1* ratio via excess LIM domain expression and consequent disturbance of endogenous *zyx-1* homeostasis, similar to that observed previously upon LIM domain overexpression in Vero cells [52]. Conversely, *azyx-1* could harbor a hitherto unknown functional domain, which upon overexpression causes loss of muscular integrity. Whether AZYX-1 *trans* effects involve other players than ZYX-1, and how these all causally fit together in regulating neuromuscular physiology and function, remains to be detailed. One observation that we cannot yet explain, is the difference in burrowing readout for the 2 *azyx-1* deletion alleles, despite their identical results for quantification of AZYX-1 loss and ZYX-1 long increase (S4 Fig). This suggests not everything about this locus is yet known. Given the conservation of LIM domain proteins across eukaryotes, and the evidence of putative syntenic conservation for *azyx-1* (Fig 1), we believe this locus provides an interesting model context for future research on LIM domain proteins and prevalence of similar overlapping genes in eukaryotic systems.

## Materials and methods

### Reagents and tools table in Table 1

**Table 1. Reagents and tools table.**

| Reagent/resource | Reference or source | Identifier or catalog number |
| --- | --- | --- |
| **Experimental models** | | |
| N2 Bristol | *Caenorhabditis* Genetics Center, University of Minnesota, USA | N/A |
| VC299 | *Caenorhabditis* Genetics Center, University of Minnesota, USA | N/A |
| NM4325 | Gift of Professor Michael Nonet, Washington University Medical School, USA | N/A |
| NM4031 | Gift of Professor Michael Nonet, Washington University Medical School, USA | N/A |

*(Continued)*

**Table 1.** (Continued)

| Reagent/resource | Reference or source | Identifier or catalog number |
|---|---|---|
| RW1596 | Gift of Professor Bart Braeckman, Ghent University, Belgium | N/A |
| PHX7030 | SunyBiotech, China | N/A |
| LSC1898 | This study | N/A |
| LSC1950 | | N/A |
| LSC1951 | | N/A |
| LSC1959 | | N/A |
| LSC1870 | | N/A |
| LSC1960 | | N/A |
| LSC1998 | | N/A |
| LSC1999 | | N/A |
| LSC2000 | | N/A |
| LSC2001 | | N/A |
| LSC2052 | | N/A |
| LSC2053 | | N/A |
| LSC2055 | | N/A |
| **Recombinant DNA** | | |
| *azyx-1p*::*azyx-1*::mNeonGreen::UTR | This study | N/A |
| *azyx-1p*::*azyx-1(gBlock)*::UTR | This study | N/A |
| *unc47p*::*mCherry* | This study | N/A |
| *unc133p*::*mCherry* | This study | N/A |
| *myo2p*::*mCherry* | This study | N/A |
| **Oligonucleotides and sequence-based reagents** | This study | S2 Table |
| **Chemicals, enzymes and other reagents** | | |
| Spike-in peptide EAVSEILETSR | GL Biochem | N/A |
| Modified Trypsin | Promega | V5111 |
| Formic Acid | Pierce | 85178 |
| Acetonitrile | Sigma/Merck | 100029 |
| Dimethyl sulfoxide | Sigma | 276855 |
| Pluronic F-127 | Sigma | P2443 |
| SspI Fast Digest | Thermo Scientific | FD0774 |
| NEBuilder HiFi DNA assembly | New England Biolabs | E2621 |
| Fluorodeoxyuridine | Sigma | F0503 |
| Dithiothreitol | Serva | 20710 |
| Iodoacetamide | Serva | 26710 |
| Urea | Sigma | U5378 |
| Thiourea | Sigma | 88810 |
| HEPES | Sigma | H3375 |
| **Software** | | |
| MaxQuant (v1.6.17.0) with Andromeda | [53] | N/A |
| Skyline (version 64-bit 21.1.0.278, 22.2.0.527) | [54] | N/A |
| Prosit | [55] | N/A |
| MSStats | [56] | N/A |
| Fiji | https://imagej.net/software/fiji | N/A |
| Blinder | [57] | N/A |
| Rstudio | https://www.rstudio.com/ | N/A |
| Inkscape | https://inkscape.org | N/A |

(*Continued*)

**Table 1.** (Continued)

| Reagent/resource | Reference or source | Identifier or catalog number |
|---|---|---|
| Affinity Designer | https://affinity.serif.com | N/A |
| **Other** | | |
| iRT Kit | Biognosys | N/A |
| Wizard Genomic DNA purification kit | Promega | A1120 |
| COPAS | Union Biometrica | |
| Pierce C18 spin columns | Thermo Scientific | 89870 |
| UltiMate 3000 RSLCnano liquid chromatography system | Thermo Fisher Scientific | N/A |
| Q-Exactive HF-X Hybrid Quadrupole-Orbitrap Mass Spectrometer | Thermo Fisher Scientific | N/A |
| sssReproSil Gold C18-AQ, 3 μm, 450 mm × 75 μm | Dr. Maisch Gmbh | r13.aq.0003 |
| ReproSil-pur C18-AQ, 5 μm, 20 mm × 75 μm | Dr. Maisch Gmbh | r15.aq.0001 |

## Worm culture

All strains used in this study (S1 Table) were cultured at 20°C on nematode growth medium (NGM) plates seeded with *E. coli* OP50 [58,59].

## Molecular biology

For *azyx-1* localization, 3,474 bp upstream of the *azyx-1* stop codon were amplified by PCR from wild-type genomic DNA, along with 558 bp of *azyx-1* 3′ UTR and fused to 5′ and 3′ ends of mNeonGreen (minus its start-AUG) using HiFi DNA assembly (NEBuilder). The resultant linear transgene was purified (Wizard Genomic DNA purification kit, Promega) and confirmed by sequencing (oligos p001-p006, S3 Table), and injected into wild-type N2 to generate *C. elegans* strain LSC1959 (see "Transgenesis" and S1 Table). For overexpression and rescue strains (LSC1950, LSC1951, LSC1960, LSC1997, LSC1999, LSC2001, LSC2052, LSC2053, LSC2055; S1 Table), a 757 bp promoter region upstream of *azyx-1* was amplified, as were 535 bp of its 3′ UTR. Next, these were fused to 5′ and 3′ ends of a 587 bp synthesized *azyx-1* gBlock (Integrated DNA Technologies (IDT); containing all *azyx-1* exons and its first intron, with the ATG at the *zyx-1a* start mutated to CTG) using HiFi DNA assembly (NEBuilder) and confirmed by sequencing (oligos p007-p010, S3 Table). To build the neuronal marker transgene, mCherry was fused to 1,800 bp of the *unc-47* promoter region and 497 bp of the *unc-47* 3′ UTR using HiFi DNA assembly (NEBuilder) and confirmed by sequencing (oligos p011-p016, S3 Table).

## CRISPR/Cas9-mediated knockout of *azyx-1a*

For the generation of the *lst1687* allele, which contains a 27 bp deletion at the beginning of the *azyx-1* ORF (positions -182 to -155 upstream of the *zyx-1a* start site), the *dpy-10* co-CRISPR strategy was used with homology-directed repair (HDR) according to Paix and colleagues [60]. The injection mix comprised 2.5 μl of recombinant codon-optimized Cas9 enzyme, 2.5 μl tracrRNA (0.17 mol/l, IDT), 1 μl *dpy-10* crRNA (0.6 nmol/μl, IDT), 1 μl of *azyx-1* crRNA (0.6 nmol/μl, IDT, S3 Table), 1 μl *dpy-10* repair template (0.5 mg/ml, Merck), and 1 μl repair template for *azyx-1* containing a 27 bp deletion that encompasses *azyx-1a* start codon (1 mg/ml, IDT, S3 Table). The mix was micro-injected in the gonads of young adult N2 Bristol wild types (Zeiss Axio Observer A1 with Eppendorf Femtojet and Eppendorf Injectman NI2) [61].

Offspring were screened for the desired CRISPR/Cas9-mediated deletion by SspI (FastDigest Thermo Fisher) cleavage of PCR products of the *azyx-1* locus, which is only possible after HDR (oligos p007 and p017, S3 Table).

The *syb7030* allele was acquired commercially (SunyBiotech) and corresponds to a precise nucleotide exchange of the *azyx-1a* start codon, wherein the -184th to -182nd basepairs upstream of *zyx-1a* were mutated from ATG to TAC. The presence of homozygous *lst1687* and *syb7030* alleles in the *azyx-1a* mutant strains LSC1898 and PHX7030 (S1 Table) was confirmed via sequencing.

## Sample collection and preparation for proteomics

Adult worms were synchronized by standard hypochlorite treatment [62]. After overnight incubation in S-basal (5.85 g NaCl, 1 g K$_2$ HPO$_4$, 6 g KH$_2$PO$_4$ in 1 L milliQ) on a rotor at 20˚C, the L1 arrested animals were grown on NGM plates seeded with *E. coli* OP50. For wild-type sampling at different ages, we collected worms at larval (L4, 48h post L1 refeeding), day 1 adult (20 h post L4 harvest) and post-reproductive, day 8 of adulthood stages. For day 8 samples, offspring were avoided by supplementing the worm cultures with 50 µl of a 50 µm fluoro-deoxyuridine (FUDR) solution every 48 h, as of the L4 larval stage (i.e., L4, and days 2, 4, and 6 of adulthood) [63]. For comparisons of wild types with *azyx-1* deletion mutants, both strains were synchronized and then grown until larval stage L4 or day 1 adult stage. For sampling, worms were washed off NGM plates with S-basal and allowed to settle in conical tubes for 10 min. Following that, the supernatant was removed and worm pellets were diluted to 15 ml in S-basal for sorting. Worms were sorted using a Complex Object Parametric Analyzer and Sorter (COPAS) platform (Union Biometrica, Holliston, Massachusetts, United States of America) for each sample individually. Four independently grown populations of worms were used per condition. We collected 200 animals per sample for day 8 or 1,000 animals per sample for all other conditions into a 1.5 ml Eppendorf LoBind tube. The worms were pelleted by spinning at 1,500 *g* for 1 min, S-basal was removed, and 200 µl of 50 mM HEPES were added to the worm pellet, spun at 1,500 *g* for 1 min and the supernatant was carefully discarded ensuring no worms were lost in pipetting. Finally, the pellet was supplemented with 1 fmol/worm of synthetic spike-in peptide (EAVSEILETSRVSGWRLFKKIS), comprising a proteotypic peptide for quantitation (EAVSEILETSR) (Vandemoortele and colleagues [64]) fused to a HiBit Tag (VSGWRLFKKIS) via a tryptic cleavage site, from a stock solution in water at a concentration of 100 fmol/µl. The pellet was snap frozen in liquid nitrogen and stored at -80˚C until further processing. The duration from initial worm collection off NGM until snap freezing was approximately 20 min and carried out at 20˚C.

For protein extraction, worm pellets were thawed on ice with 100 µl of lysis buffer (8 M Urea, 2 M Thiourea in 10 mM HEPES) and lysed by sonication using a probe sonicator (40% amplitude, 5 s ON, 10 s OFF × 10). The lysate was spun at 15,000 *g* for 10 min and the supernatant was transferred to a fresh 1.5 ml Eppendorf LoBind tube. Protein concentration was estimated using a Bradford assay and sample aliquots corresponding to 50 µg of total protein were processed further for LC-MS/MS. For this, each sample was reduced with 5 mM dithiothreitol at 56˚C for 30 min and alkylated with 25 mM of iodoacetamide for 20 min at room temperature. The lysate was diluted to 1M urea and digested overnight at 37˚C with 2 µg of sequencing-grade trypsin (Promega), after which the sample was acidified to 0.1% formic acid, cleaned using Pierce C18 spin columns as per the manufacturer's protocol and dried in a Savant SpeedVac. The dried peptides were dissolved to 0.1 µg/µl in 2% acetonitrile/98% H$_2$O/0.1% formic acid (FA)/0.1X Biognosys iRT peptides (for retention time calibration).

## Peptide ion selection for targeted quantification

Peptide ions useful for quantification of proteins of interest (ZYX-1 and AZYX-1), and of proteins used for data normalization (GPD-3, HIS-24, spike-in) were selected based on an unscheduled parallel reaction monitoring (PRM) experiment. To accurately normalize data across age and conditions, we chose 3 normalization options: 2 relying on endogenous *C. elegans* proteins—*viz*. GPD-3 (GAPDH homolog—4 peptides), HIS-24 (Histone homolog—4 peptides)—and one relying on the externally added synthetic spike-in peptide (1 peptide). Skyline-daily was used to build an initial library [54]. For all proteins of interest, all theoretically predicted tryptic peptides with a length between 7 and 26 amino acids were added to the initial spectral library. In total, 98 peptide precursor ions were selected and measured in an unscheduled PRM experiment that was run on a pooled sample consisting of all peptide samples used in this study and analyzed with Skyline. Subsequently, 23 measured peptide precursors representing 22 peptides and 5 target proteins (ZYX-1, AZYX-1, GPD-3, HIS-24, Spike-in) were selected for the final PRM measurements. Additionally, 11 MS1 ions of the Biognosys iRT reference peptides were included in the precursor list. Details of all peptides and corresponding protein(s), including their uniqueness in the proteome database, can be found in S2 Table.

## Targeted LC-MS/MS measurements

Targeted measurements using scheduled PRM were performed with a 50-min linear gradient on a Dionex UltiMate 3000 RSLCnano system coupled to a Q-Exactive HF-X mass spectrometer (Thermo Fisher Scientific). The spectrometer was operated in PRM and positive ionization mode. MS1 spectra (360 to 1,300 m/z) were recorded at a resolution of 60,000 using an AGC target value of $3 \times 10^6$ and a MaxIT of 100 ms. Targeted MS2 spectra were acquired at 60,000 resolution with a fixed first mass of 100 m/z, after HCD with a normalized collision energy of 26%, and using an AGC target value of $1 \times 10^6$, a MaxIT of 118 ms and an isolation window of 0.9 m/z. In a single PRM measurement, 23 + 11 MS1 peptide ions (see above) were targeted with a 5-min scheduled retention time window. The cycle time was approximately 2.1 s, which leads to about 10 data points per chromatographic peak.

## Targeted mass spectrometric data analysis

PRM data were analyzed using Skyline (version 64-bit 21.1.0.278 and 22.2.0.527) [54]. Peak integration, transition interferences, and integration boundaries were reviewed manually, considering 4 to 6 transitions per peptide. To discriminate true from false positive peptide detection, filtering according to correlation of PRM fragment ion intensities was carried out. For this purpose, an experimental spectral library was built from the PRM data itself, by searching these with MaxQuant and then loading the generated search results back into Skyline. For confident peptide identification, a "Library Dot Product" $\geq 0.85$, as well as a mass accuracy $\leq 10$ ppm ("Average Mass Error PPM") were required. We also manually verified the correlation between PRM fragment ion intensties and spectra predicted with the artifical intelligence algorithm Prosit [55]. For peptide and protein quantification, chromatographic peak areas were exported from Skyline in MSStats format, and further processing, quantification, statistical analysis, and visualization were performed in RStudio with the MSStats package [56]. For HIS-24, 3 most consistent peptides out of 4 were considered for downstream analysis and peptide FISQNYK was omitted. The data were $\log_2$ transformed, processed as per default MSStats parameters, and visualized using the ggplot2 package of *R*. For age analysis, data were normalized to spike-in peptide (1 fmol/worm) and L4 samples were used as the reference. For *azyx-1a* mutant and wild-type comparison, all 3 normalizations were considered (i.e., GPD-3, HIS-24, and spike-in). To evaluate the internal control stability the pair-wise ratios of each

combination were calculated based on Vandesompele and colleagues [65] and the equality of variance was evaluated using Levene's test. The mass spectrometric raw files acquired in PRM mode and the Skyline analysis files have been deposited to Panorama Public (Sharma and colleagues [66]) and can be accessed via https://panoramaweb.org/azyx1.url.

## Transgenesis

For in vivo localization of *azyx-1*, the *lstEx1065* extra-chromosomal array was generated by mixing 25 ng/µl purified linear transgene [*azyx-1p*::*azyx-1*+mNeonGreen::*azyx-1* 3′ UTR] (see "Molecular biology") with 25 ng/µl coelomocyte-restricted co-injection marker [*unc-122p*::*DsRed*] and 50 ng/µl 1-kb ladder (Thermo Scientific) as carrier DNA. This was injected into Bristol wild type (N2) to generate LSC1959 (S1 Table).

All genetic overexpressions and rescues of *azyx-1* strains were created via injection of 25 ng/µl of linear transgene [*azyx-1p*::*azyx-1*(gBlock)::*azyx-1* 3′ UTR] with 12.5 ng/µl pharyngeal co-injection marker [*myo-2p*::mCherry] and 50 ng/µl of 1-kb ladder (Thermo Scientific) as carrier DNA unless otherwise noted (S1 Table). For strains transgenically expressing the *unc-47p*::mCherry neuronal marker (LSC1998 and LSC1999; S1 Table), 10 ng/µl of this linear construct was injected along with 5 ng/µl pharyngeal co-injection marker [*myo-2p*::mCherry] and 50 ng/µl of 1-kb ladder (Thermo Scientific), with the addition of 10 ng/µl overexpression transgene [*azyx-1p*::*azyx-1*(gBlock):: *azyx-1* 3′ UTR] for LSC1999.

Transgenic strains were always confirmed by observation of co-injection marker presence via fluoresence microscopy, followed by PCR and sequencing of the added target sequences. For overexpression of *azyx-1* in the *zyx-1* reporter background (LSC1870, which expresses mCherry::*zyx-1*;*zyx-1*::GFP, S1 Table), the extrachromosomal array (NM3425) was integrated with UV irradiation as per [67] and outcrossed twice with wild type (N2 Bristol). All injections using a Zeiss Axio Observer A1 with Eppendorf Femtojet and Eppendorf Injectman NI2 were performed targeting syncytial gonads of young adults.

## Confocal imaging

Confocal microscopy was performed using either an Olympus FluoView 1000 (IX81) or a Zeiss LSM900 confocal microscope. To obtain synchronized L4 larvae, a timed egg-laying was performed 48 h before imaging, whereas day 1 adults were synchronized by picking L4 larvae approximately 16 h before imaging. Worms were anaesthetized using of 500 mM sodium azide and mounted on 2% agarose pads. Using Fiji [68], resulting z-stacks were analyzed by performing a sum of slices projection and selecting the region of interest (worm or neuron) with the polygon selection tool. In this region of interest, the mean pixel intensity was measured.

## Manual scoring of muscular filaments

Day 1 adults were imaged with confocal microscopy to investigate muscle fiber integrity in control (RW1595, *n = 75*), overexpressor (LSC2000, *n = 52*; LSC2055 *n = 40*) and mutant strains (LSC2001, *n = 57*). The resulting image files were randomized in Blinder freeware (Cothren and colleagues), visually assessed and scored between 2 qualitative classes of muscle filaments: (1) normal well-organized or (2) mildly damaged or disorganized (Fig 5A), based on previously reported manual scoring parameters [69]. Briefly: muscle fibers displaying tightly organized filaments aligned in a parallel manner were classified as normal. The increased presence of breakage, thinning, or disorganization of individual muscle filaments were classified as disorganized. Fisher's exact test was used to determine association between strains and muscle integrity. Bonferroni corrections were used for multiple comparisons and adjusted *p*-values were reported.

## Burrowing assay

Burrowing assays were performed with synchronized day 1 adults and executed essentially as described by Lesanpezeshki and colleagues [32], with minor adjustments. Briefly, 20 gravid adults for each replicate were allowed to lay eggs on a seeded NGM plate for 3 h and subsequently removed while allowing the eggs to hatch and grow at 20˚C. After 70 h, worms were washed off the NGM plates with S-basal and transferred onto unseeded NGM plates to induce a starvation response. After 1 h, 30 adult worms were dropped in 50 μl 26% w/v Pluronic F-127 in a well of a Corning Costar 12-well plate and covered with 2.5 ml of 26% w/v Pluronic F-127 (Sigma-Aldrich) at 14˚C. After 15 min at 20˚C, the Pluronic F-127 had gelated and a droplet of 20 μl 100 mg/ml *E. coli* HB101 was added on top as a chemoattractant (time = 0). The bacterial droplet was monitored every 30 min for 3 h to calculate a chemotaxis index as the percentage of worms that had cumulatively reached the bacteria (out of the 100% corresponding to $n \geq 30$). At each time point of observation, worms that had reached the bacterial pellet were removed to avoid crowding and reburrowing. Statistical significance was determined by two-way ANOVA.

## Supporting information

**S1 Fig. *azyx-1* is expressed consistently from larval to adult stages.** Localization of *azyx-1p::azyx-1+mNeonGreen::UTR* in (**A**) body wall (scale bar, 100 μm), (**B**) head, (**C**) vulval muscle, and (**D**) tail (scale bar, 20 μm) of L4 stage worms.
(TIF)

**S2 Fig. AZYX-1 and ZYX-1 proteoform abundance changes from larval to reproductive and post-reproductive adulthood.** Normalized intensity of transition ions for peptides corresponding to (**A**) AZYX-1 (7 peptides) and (**B**) ZYX-1 (6 peptides) measured at L4 (orange), day 1 (green) and day 8 (blue) of adulthood ($n$ = 4) with mean intensity across replicates (horizontal black line). Data normalized to spike-in (1 fmol/worm). Formal statistical analysis of data was performed and indicated in main text corresponding to Fig 3A–3D. Data used to generate figures can be found in S1 Data.
(TIF)

**S3 Fig. Robustness across normalization options confirms observed *cis* regulation of *zyx-1* by *azyx-1*.** Fold change and standard error of individual ZYX-1 peptides in *azyx-1* mutant vs. *wt* with data normalized to (**A**) spike-in peptide or (**B**) HIS-24. (**C**) Fold change and standard error of the only detectable AZYX-1 peptide in *azyx-1* mutant strain (LSC1898), likely corresponding to AZYX-1b isoform, upon AZYX-1a start codon deletion with a significant downregulation in comparison to *wt* at day 1 of adulthood (normalized to GPD-3, *p = 0.0017*). (**D**) Distribution of raw intensity of transition ions with mean and standard error for spike-in (1 peptide), GPD-3 (4 peptides), and HIS-24 (3 peptides) across 4 (colored) biological replicates. None of the normalization methods differed significantly from the others (Levene's test *p = 0.991* for normalized ratios of raw intensities, with pairwise *p*-values vs. spike-in *0.99*, vs. GPD-3 *0.72*, and vs. HIS-24 *0.72*). Data used to generate figures can be found in S1 Data.
(TIF)

**S4 Fig. Robustness of observed *cis* regulation of *zyx-1* by *azyx-1* across AZYX-1a start codon deletion mutants.** Fold change and standard error of peptides in *azyx-1* mutant strains (Δ27bp and ATG -> TAC) corresponding to (**A**) ZYX-1, where ZYX-1.1–1.3 peptides are specific to long zyxin isoforms and ZYX-1.5 being the only detectable peptide shared between all isoforms, and (**B**) the only detectable AZYX-1 peptides likely corresponding to AZYX-1b

isoform; *p*-value: *** <0.001, *n* = 5 biological replicates, L4 larval stage, with data normalized to GPD-3. Data used to generate figures can be found in S1 Data.
(TIF)

**S5 Fig. *azyx-1* affects burrowing behavior.** Chemotaxis index of burrowing assay in day 1 adults, as cumulatively observed over 180 min for the *azyx-1* (Δ27bp) mutant (LSC1898), rescue (LSC1951), and overexpressor (LSC1950) strains in comparison to positive (*zyx-1(gk190)*) and negative (wild type) controls, *n* = 30 per condition in 5 biologically independent replicates (30 × 5); two-way ANOVA *p = 0.0025* for strain and time. Tukey HSD wild type vs. *azyx-1* OE *p = 0.047*, vs. *azyx-1* (Δ27bp) *p = 0.018*, vs. *zyx-1 p = 0.002*. Data used to generate figure can be found in S1 Data.
(TIF)

**S1 Table. Strains used in this study, with strain name, genotype, and source.**
(XLSX)

**S2 Table. Transition list for PRM experiment targeting spike-in, GPD-3, HIS-24, ZYX-1, AZYX-1, and Biognoysis iRT peptides with their m/z, charge state (polarity), LC retention time start and end, normalized collision energy (NCE), peptide sequence, protein name, uniqueness/commonness of the peptide within *C. elegans* proteome.**
(XLSX)

**S3 Table. Oligos used in this study, with direction/type, nucleotide sequence, and assigned name.**
(XLSX)

**S4 Table. Predicted uORF, uoORF, and oORF with similar synteny as *azyx-1* in *zyx-1* orthologs across 8 species.** For each, OpenProt ID, species, protein length (# amino acids), molecular weight (kDa), isoelectric point, transcript ID, (localization vs.) associated LIM gene, sequence, and ORF type are provided.
(XLSX)

**S1 Data. Spreadsheet containing underlying data for Figs 3–3D, 4B–4D, 4G, 5B, 5C, S2, S3, S4 and S5.**
(XLSX)

## Acknowledgments

We would like to thank Prof. Kathrin Gieseler (Université Claude Bernard Lyon 1, France), Prof. Michael Nonet (Washington University in St. Louis, USA) and Prof. Bart Braeckman (UGent, Belgium) for providing strains, and Dr. Marlies Peeters and Dr. Gerben Menschaert (UGent, Belgium) for valuable discussions.

## Author Contributions

**Conceptualization:** Bhavesh S. Parmar, Liesbet Temmerman.

**Data curation:** Bhavesh S. Parmar.

**Formal analysis:** Bhavesh S. Parmar, Amanda Kieswetter, Ellen Geens, Christina Ludwig.

**Funding acquisition:** Bhavesh S. Parmar, Liesbet Temmerman.

**Investigation:** Bhavesh S. Parmar.

**Methodology:** Bhavesh S. Parmar, Amanda Kieswetter, Elke Vandewyer, Christina Ludwig.

**Project administration:** Liesbet Temmerman.

**Resources:** Amanda Kieswetter, Elke Vandewyer, Christina Ludwig.

**Supervision:** Liesbet Temmerman.

**Validation:** Bhavesh S. Parmar.

**Visualization:** Bhavesh S. Parmar, Amanda Kieswetter.

**Writing – original draft:** Bhavesh S. Parmar.

**Writing – review & editing:** Amanda Kieswetter, Ellen Geens, Christina Ludwig, Liesbet Temmerman.

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
