## [Editor Report · Decision Letter 0]

19 Sep 2022

Dear Dr Temmerman, 

Thank you for submitting your manuscript entitled "A novel overlapping gene azyx-1 affects the translation of zyxin in C. elegans" for consideration as a Research Article by PLOS Biology.

Your manuscript has now been evaluated by the PLOS Biology editorial staff, as well as by an academic editor with relevant expertise, and I am writing to let you know that we would like to pursue your manuscript further and invite you to submit a revised version of the manuscript.

However, before we can invite a revision, we need you to complete your submission by providing the metadata that is required for full assessment. To this end, please login to Editorial Manager where you will find the paper in the 'Submissions Needing Revisions' folder on your homepage. Please click 'Revise Submission' from the Action Links and complete all additional questions in the submission questionnaire.

To provide the metadata for your submission, please Login to Editorial Manager (https://www.editorialmanager.com/pbiology) within two working days, i.e. by Sep 21 2022 11:59PM.

Kind regards,

Richard

Richard Hodge, PhD

Associate Editor, PLOS Biology

rhodge@plos.org

PLOS

---

## [Editor Report · Decision Letter 1]

21 Sep 2022

Dear Dr Temmerman,

Thank you very much for submitting your manuscript "A novel overlapping gene azyx-1 affects the translation of zyxin in C. elegans" for consideration as a Research Article at PLOS Biology. As you know, your manuscript and plan of revision have been evaluated by the PLOS Biology editors and by an Academic Editor with relevant expertise. I have also included some specific comments from the Academic Editor below my signature. 

Based on your responses to the reviews from Reviews Commons, we would welcome re-submission of a revised version that takes into account the reviewers' comments. In addition, the Academic Editor agrees with the concerns raised by Reviewer #2 regarding the 27 bp azyx-1 deletion allele and notes that additional data using a ATG start codon mutation should be included in the revised manuscript.

We cannot make any decision about publication until we have seen the revised manuscript and your response to the reviewers' comments. Your revised manuscript is also likely to be sent for further evaluation by the original reviewers.

We expect to receive your revised manuscript within 3 months. Please email us (plosbiology@plos.org) if you have any questions or concerns, or would like to request an extension. At this stage, your manuscript remains formally under active consideration at our journal; please notify us by email if you do not intend to submit a revision so that we may end consideration of the manuscript at PLOS Biology.

**IMPORTANT - SUBMITTING YOUR REVISION**

*Re-submission Checklist*

*Published Peer Review*

*PLOS Data Policy*

*Blot and Gel Data Policy*

Sincerely,

Richard

Richard Hodge, PhD

Associate Editor, PLOS Biology

rhodge@plos.org

COMMENTS FROM ACADEMIC EDITOR

The response to the 27bp deletion allele is not fully satisfying. I side with reviewer 2 here. Mutation of the ATG would be a better way. One can still argue that mutation of the ATG would affect some binding site of a transcription factor, chances for that are much lower than a 27bp deletion close to the start of a gene. One could even mutate the ATG in different ways to further reduce those chances.

---

## [Decision Letter · Decision Letter 2]

14 Jul 2023

Dear Dr Temmerman,

Thank you for your patience while we considered your revised manuscript "A novel overlapping gene azyx-1 affects the translation of zyxin in C. elegans" for publication as a Research Article at PLOS Biology. This revised version of your manuscript has been evaluated by the PLOS Biology editors, the Academic Editor and the original reviewers.

Based on the reviews, I am pleased to say that we are likely to accept this manuscript for publication, provided you satisfactorily address the following data and other policy-related requests that I have provided below (A-E):

(A) We would like to suggest the following modification to the title: 

“"azyx-1 is a new gene that overlaps with zyxin and affects its translation in C. elegans thereby impacting muscular integrity and locomotion"

(B) You may be aware of the PLOS Data Policy, which requires that all data be made available without restriction: http://journals.plos.org/plosbiology/s/data-availability. For more information, please also see this editorial: http://dx.doi.org/10.1371/journal.pbio.1001797

-Supplementary files (e.g., excel). Please ensure that all data files are uploaded as 'Supporting Information' and are invariably referred to (in the manuscript, figure legends, and the Description field when uploading your files) using the following format verbatim: S1 Data, S2 Data, etc. Multiple panels of a single or even several figures can be included as multiple sheets in one excel file that is saved using exactly the following convention: S1_Data.xlsx (using an underscore).

-Deposition in a publicly available repository. Please also provide the accession code or a reviewer link so that we may view your data before publication. 

Figure 3A-D, 4B-D, 4G, 5B-C, S2A-B, S3A-D, S4A-B, S5

(C) Thank you for already depositing the mass spectrometry data in a pubic repository (Panorama Public). However, I note that when I clicked on the DOI link, there was an error with the URL. The associated ProteomeXchange deposition (PXD034878) also does not appear to be publicly released. I would be grateful if you could please make the ProteomeXchange deposition publicly available at this stage and ensure that the DOI for the Panorama Public deposition is correct/activated. 

(D) Please also ensure that each of the relevant figure legends in your manuscript include information on *WHERE THE UNDERLYING DATA CAN BE FOUND*, and ensure your supplemental data file/s has a legend.

(E) Please ensure that your Data Statement in the submission system accurately describes where your data can be found and is in final format, as it will be published as written there. This includes adding the DOI/accession numbers of the deposition for the mass spectrometry data. 

We expect to receive your revised manuscript within two weeks. 

*Published Peer Review History*

*Press*

Sincerely,

Richard

Richard Hodge, PhD

rhodge@plos.org

Reviewer remarks:

Reviewer #1: The authors have addressed all of my comments and the revised manuscript remains an important contribution to the field(s).

Reviewer #2 (Erin J Cram, signs review): The authors have addressed my concerns. 

Reviewer #3: I am happy with the improvements made to the manuscript. The generation of the sb7030 allele gives confidence to the findings arising from the other loss-of-function allele. The authors have adequately dealt with my concerns from the initial review.

---

## [Editor Report · Decision Letter 3]

16 Aug 2023

Dear Dr Temmerman,

Thank you for the submission of your revised Research Article "azyx-1 is a new gene that overlaps with zyxin and affects its translation in C. elegans, impacting muscular integrity and locomotion." for publication in PLOS Biology. On behalf of my colleagues and the Academic Editor, Rene Ketting, I am pleased to say that we can accept your manuscript for publication, provided you address any remaining formatting and reporting issues. These will be detailed in an email you should receive within 2-3 business days from our colleagues in the journal operations team; no action is required from you until then. Please note that we will not be able to formally accept your manuscript and schedule it for publication until you have completed any requested changes.

PRESS

Sincerely, 

Richard

Richard Hodge, PhD

rhodge@plos.org

PLOS
